# Achieving Minimum Standards for Infection Prevention and Control in Sierra Leone: Urgent Need for a Quantum Leap in Progress in the COVID-19 Era!

**DOI:** 10.3390/ijerph19095642

**Published:** 2022-05-06

**Authors:** Bobson Derrick Fofanah, Arpine Abrahamyan, Anna Maruta, Christiana Kallon, Pruthu Thekkur, Ibrahim Franklyn Kamara, Charles Kuria Njuguna, James Sylvester Squire, Joseph Sam Kanu, Abdulai Jawo Bah, Sulaiman Lakoh, Dauda Kamara, Veerle Hermans, Rony Zachariah

**Affiliations:** 1World Health Organization Country Office, 21A-B Riverside Drive, Brookfields, Freetown 00232, Sierra Leone; marutaa@who.int (A.M.); ikamara@who.int (I.F.K.); njugunach@who.int (C.K.N.); 2Tuberculosis Research and Prevention Center (TBRPC), Yerevan 0014, Armenia; arpine_abrahamyan@yahoo.com; 3National Infection Prevention and Control Coordinating Unit, Ministry of Health and Sanitation, Freetown 00232, Sierra Leone; christy.conteh@yahoo.com; 4Centre for Operational Research, International Union against Tuberculosis and Lung Disease, 75001 Paris, France; pruthu.tk@theunion.org; 5Directorate of Health Security and Emergencies, Ministry of Health and Sanitation, Freetown 00232, Sierra Leone; jmssquire@yahoo.com (J.S.S.); samjokanu@yahoo.com (J.S.K.); 6College of Medicine and Allied Health Sciences, University of Sierra Leone, Freetown 00232, Sierra Leone; abdulaijawobah@yahoo.com (A.J.B.); lakoh2009@gmail.com (S.L.); 7Institute of Global Health and Development, Queen Margaret University, Edinburgh EH21 6UU, UK; 8Department of Medicine, University of Sierra Leone Teaching Hospitals Complex, Freetown 00232, Sierra Leone; 9Water, Sanitation, and Hygiene (WASH) Program, Ministry of Health and Sanitation, Freetown 00232, Sierra Leone; daudakamara50@gmail.com; 10Médecins Sans Frontières, Operational Centre Brussels, 68, Rue de Gasperich, 1617 Luxembourg, Belgium; veerle.hermans@luxembourg.msf.org; 11UNICEF, UNDP, World Bank, WHO Special Programme for Research and Training in Tropical Diseases (TDR), Avenue Appia 20, 1211 Geneva, Switzerland; zachariahr@who.int

**Keywords:** SORT IT, operational research, WASH, universal health coverage, IPCAT, IPCAF, Ebola, health care-associated infection, IPC programme

## Abstract

Introduction: Good Infection prevention and control (IPC) is vital for tackling antimicrobial resistance and limiting health care-associated infections. We compared IPC performance before (2019) and during the COVID-19 (2021) era at the national IPC unit and all regional (4) and district hospitals (8) in Sierra Leone. Methods: Cross-sectional assessments using standardized World Health Organizations IPC checklists. IPC performance scores were graded as inadequate = 0–25%, basic = 25.1–50%, intermediate = 50.1–75%, and advanced = 75.1–100%. Results: Overall performance improved from ‘basic’ to ‘intermediate’ at the national IPC unit (41% in 2019 to 58% in 2021) and at regional hospitals (37% in 2019 to 54% in 2021) but remained ‘basic’ at district hospitals (37% in 2019 to 50% in 2021). Priority gaps at the national IPC unit included lack of: a dedicated IPC budget, monitoring the effectiveness of IPC trainings and health care-associated infection surveillance. Gaps at hospitals included no assessment of hospital staffing needs, inadequate infrastructure for IPC and lack of a well-defined monitoring plan with clear goals, targets and activities. Conclusion: Although there is encouraging progress in IPC performance, it is slower than desired in light of the COVID-19 pandemic. There is urgent need to mobilize political will, leadership and resources and make a quantum leap forward.

## 1. Introduction

Infection prevention and control (IPC) at health facilities is a scientific approach and practical solution to prevent infection and harm to patients and health workers [1]. IPC is a central pillar of health system resilience and contributes to the quality of universal health coverage, since it is relevant to health workers, patients and visitors at every health-care encounter [2]. Importantly, it is fundamental to reducing health care-associated infections (HAIs) [3].

In general, many commonly used IPC measures are resource-intensive, including the use of single-occupancy hospital isolation rooms and single-use personal protective equipment (PPE) [4]. In resource-constrained settings (LMICs), limited availability of these resources and limited staffing may pose challenges. Lack of adequate training, limited microbiological services, and other competing health priorities (e.g., COVID-19) may compromise the implementation of effective IPC programme [5]. In LMICs, poor sanitation, lack of health literacy and resource shortages are just few of the many obstacles that confront when tasked with infectious disease outbreaks.

A systematic review assessing HAIs in Low- and Middle-Income Countries (LMICS) reported a pooled prevalence of 15.5 per 100 patients which is about three-fold higher than in high-income countries [5]. Such infections result in prolonged hospital stays, high costs to patients, and avoidable deaths of patients and health workers [6]. The high prevalence of HAIs in LMICS is a ‘proxy’ of inadequate IPC that needs to be strengthened. A systematic review reported that about 35% to 55% of HAIs can be averted through multifaceted interventions aimed at controlling and preventing infections [7]. IPC is also one of the strategic priorities of the global and national action plans for tackling antimicrobial resistance (AMR) [8]. The logic being that ‘every infection prevented is an antibiotic treatment avoided’ [9].

In countries such as Sierra Leone, achieving minimum IPC standards is paramount to prevent the transmission of infectious diseases that pose global threats, such as pandemic influenza, Ebola virus disease (EVD), other viral haemorrhagic fevers and of recent concern, COVID-19. The country was badly hit by the 2014–2015 EVD outbreak in West Africa, with over 14,000 reported cases and 3955 EVD deaths including those of 221 health workers [10,11]. The rapid transmission of EVD in the country, especially among healthcare workers, was attributed to poor IPC infrastructure and inadequate IPC practices prior to the outbreak.

Recognizing the importance of IPC, the World Health Organization (WHO) developed minimum IPC standards that should be in place in all health facilities [3]. This includes checklists to assess IPC performance at the national IPC unit (Instructions for the National Infection Prevention and Control Assessment, IPCAT) [12] and at health facility levels (Infection Prevention and Control at Facility Level, IPCAF) [13]. These checklists are closed-formatted questionnaires with an associated scoring system that allows grading of IPC into levels, namely: inadequate, basic, intermediate or advanced. They are useful for baseline assessments of IPC status, which can be followed over time by repeated assessments to document progress and facilitate improvements. Most of the published studies in the Africa region have used country specific checklists for IPC performance assessments and were conducted in pre-COVID era [14,15,16].

In Sierra Leone, there has been one study on IPC performance assessment from Kenema district, which used the Ministry of Health of Sierra Leone (MOHS) assessment checklist. The study showed that IPC compliance at the district hospital increased from 69% in 2016 to 73% in 2018 (expected minimal threshold = 70%; desired threshold ≥ 85%) [17]. There has, however, been no published study from Sierra Leone or the African region that has assessed IPC implementation at the national IPC unit and in public health facilities, using the WHO assessment checklists (IPACT and IPACF).

A country-wide IPC performance assessment of the national IPC unit and secondary/tertiary health facilities using WHO checklists was conducted in 2021 in the COVID-19 era. Since a similar IPC assessment was conducted in 2019, this provides an opportunity to assess how far we have come from the pre-COVID-19 era. Such information could help the IPC programme managers to identify the areas of concern (gaps) and catalyse action for improvement.

During the COVID-19 pandemic, there was additional emphasis on improving IPC practices [4]. However, there were no published studies globally on how IPC performance changed in the health facilities with the advent of the COVID-19 pandemic. Thus, we aimed to assess the change in the IPC scores (performance) using WHO checklists, in 2019 (pre-COVID-19 era) and 2021 (COVID-19 era) at the national IPC unit and public health facilities. We also identified the gaps in specific components of IPC implementation.

## 2. Materials and Methods

### 2.1. Study Design

This was a before (pre-COVID-2019) and after (COVID-19 era-2021) comparison of the data from the cross-sectional IPC assessments conducted routinely by the IPC programme of Sierra Leone.

### 2.2. Study Setting

#### 2.2.1. General Setting

Sierra Leone is a tropical country on the Atlantic coast of West Africa lying between Guinea and Liberia. It is divided into 16 districts with five regions, and has an estimated population of about 8 million people [18]. Public healthcare facilities are tiered into tertiary hospitals (6), regional hospitals (4), district hospitals (8), other secondary hospitals (11) and Peripheral Health Units (PHUs). The PHUs include Community Health Centres, Community Health Posts, and Maternal and Child Health Posts. The PHUs are delivery points for primary health care while hospitals deliver secondary and/or tertiary health care [19].

Sierra Leone struggled through a decade of civil war (1991 to 2002) that devastated its health infrastructure. The country has faced a series of infectious disease outbreaks of pandemic influenza, EVD, other viral haemorrhagic fevers and of recent, COVID-19. Recently, Lassa fever, a highly fatal, viral-haemorrhagic disease, is another public health threat in Sierra Leone that also requires high compliance to IPC standards. The advent of the COVID-19 pandemic and growing threats of AMR in resource-limited settings such as Sierra Leone further emphasize the importance of IPC and its monitoring.

Since Sierra Leone recorded it first case of COVID-19 in March 2020, a national COVID-19 emergency response structure (NACOVERC) and district COVID-19 emergency response centres (DECOVERC) were established with clear terms of reference to provide oversight and strategize the overall COVID-19 response activities. Members of these structures included state security and Ministry of Health personnel across all level of health sector, with IPC being one of the technical pillars within the structures.

#### 2.2.2. Specific Setting

The 2014–2015 EVD outbreak drew attention to the gaps in IPC, leading to the establishment in 2015 of a national IPC unit housed at the MOHS and supported by WHO and other public health partners. This unit is mandated to provide leadership as well as to coordinate, train and supervise the implementation and strengthening of all IPC standards countrywide. It has one national IPC coordinator and seven supporting IPC officers. This team performs routine field supervisory visits to all public health facilities and is also responsible for filling up the IPC checklists on an annual basis.

At regional hospital level, all IPC activities are coordinated by designated IPC personnel. At district level, IPC district committees and IPC supervisors are responsible for implementation of the National IPC Policy and Guidelines. Each healthcare facility has an IPC focal person. These individuals are charged with coordinating IPC activities at various levels.

The WHO country office provides technical support through an IPC support team and resource support to the national IPC unit and to health care facilities (for example the local production of alcohol-based hand-rub solutions). The IPC team from WHO works in close collaboration with the national IPC unit of Sierra Leone.

At various levels, the IPC programs have its responsible personnel/body who is accountable for action points and addressing the existing gaps in IPC. The figure below shows the structural arrangement of IPC program in Sierra Leone (Figure 1).

##### The WHO Checklists for IPC Assessments

The country uses WHO checklists for assessment of IPC standards at the national IPC unit (IPCAT) [12], and at health facilities (IPCAF) [13]. These checklists are standardized, closed-formatted questionnaires with an associated scoring system. They are developed based on the WHO guideline on “Core Components of Infection Prevention and Control at the National and Acute Health Care Facility Levels” [3].

The eight core components included in the IPCAF (for health facilities) checklist are: (1) IPC programme, (2) IPC guidelines, (3) IPC education and training, (4) HAI surveillance, (5) multimodal strategies for implementation of IPC interventions, (6) monitoring/audit of IPC practices and feedback, (7) workload, staffing and bed occupancy and (8) built environment, materials and equipment for IPC. The IPCAF checklist has a total score of 800 with eight core components and 81 indicators [13].

The IPCAT checklist (for the national IPC programme unit) includes the first six of the above-mentioned components. The IPCAT checklist has a total of 112 indicators framed into yes/no response options [12]. A percentage score is calculated based on the total number of ‘yes’ responses. For the IPCAT, a single element is either fully implemented (yes) or not (no). The IPCAF checklist is in a paper-based format and the IPCAT is available in excel format with automated formulas for calculating scores and percentages. The scale and scoring of IPCAF and IPCAT is shown in Appendix A.

### 2.3. Study Inclusion and Period

The study included data from assessment of the national IPC unit and four regional and eight district hospitals of Sierra Leone. These regional and district hospitals were located in four provincial regions of Sierra Leone, namely the Northern, Southern, Eastern, and North-Western regions. The IPC performance assessment of 2019 was conducted in the months of May and July, while the 2021 assessment was conducted in June and July.

### 2.4. Data Collection, Variables and Sources

#### 2.4.1. Data Collection and Entry

Data for the 2019 annual IPC assessment were already available in the paper-based checklists (hard-copies). Data for the 2021 annual assessment were collected by the IPC team using the same paper-based checklists and in the same routine manner as in 2019. The data from each paper-based checklist were entered into a customized excel database developed by the WHO for calculating scores of IPCAT and IPCAF per assessed facility. The calculated scores from assessment of the national IPC unit, four regional hospitals and eight district hospitals were merged into one excel database for further analysis.

#### 2.4.2. Data Variables

To assess the change in minimum IPC scores at the level of the national IPC unit between 2019 (pre-COVID-19) and 2021 (COVID-19 era), the year, the list of indicators and scores in the six core components of the IPCAT tool were included. Similarly, to compare the change in minimum IPC scores at four regional and eight district hospitals, the year, the name of the assessed facility and list of indicators and scores in the eight core components of the IPCAF tool were included.

### 2.5. Statistical Analysis

Every component of the IPCAT and IPCAF tool contributes a score of 100 with a maximum total score of 600 for IPCAT (six core components) and 800 for IPCAF (eight ore components). Based on the score obtained for each component and subcomponent, a percentage score was calculated (score obtained divided by maximum component score multiplied by 100). IPC performances in each core component and subcomponent were graded based on the obtained percentage: (i) inadequate (0–25%), (ii) basic (25.1–50%), (iii) intermediate (50.1–75%), and (iv) advanced (75.1–100%). The median scores for regional and district hospitals were calculated for summarizing the IPC scores of the four regional and eight district hospitals. The percentage change in the IPC scores between 2019 and 2021 were calculated for each core component by subtracting the percentage score of 2019 from the percentage score of 2021. Radar charts were used to depict the percentage scores in the core components at the national IPC unit, regional hospitals and district hospitals during the 2019 and 2021 assessments. IPCAT and IPCAF sub-components with inadequate scores (≤25%) were considered as gaps and were listed.

## 3. Results

### 3.1. Assessment of National IPC Unit Using IPCAT

#### 3.1.1. Change in IPC Scores from 2019 to 2021

The overall grade improved from ‘basic’ (41%) to ‘intermediate’ (58%) with ‘IPC guidelines’ reaching the ‘advanced’ level. The highest improvement was seen in ‘IPC education and training’ (44% change) while the lowest was in ‘HAI surveillance’ (8% change) and ‘monitoring/audits of IPC practices and feedback’ (5% change). The ‘multi-modal strategies’ remained at the ‘basic’ level during both assessments (Table 1 and Figure 2).

#### 3.1.2. Gaps in the IPCAT Sub-Components at National IPC Unit

The percentage scores of the IPCAT sub-components in 2019 and 2021 are presented in Table 2. In 2021, the five sub-components that reached 100% were under ‘IPC Guideline’, ‘IPC education and training’ and ‘multimodal strategies’. In 2021, the sub-components with inadequate scores (≤25%, gaps) are highlighted in red colour and include ‘monitoring of training and education’, sub-components under the ‘HAI surveillance’ and ‘multimodal strategies’. Not having a dedicated budget was one of the gaps noted under the IPC programme (IPCAT2–1.1.7).

### 3.2. Assessment of Regional and District Hospital Using IPCAF

#### 3.2.1. Change in IPC Scores in Regional Hospitals from 2019 to 2021

In the regional hospitals (N = 4), the overall grade improved from ‘basic’ (37%) to ‘intermediate’ level (54%). The ‘IPC education and training’ reached ‘advanced’ level by 2021. The highest increase in scores was seen in the ‘HAI surveillance’ component (40% change), but this still remained at the ‘basic’ level. The least increase in percentage scores was seen in the ‘workload, staffing and bed occupancy’ (5% change), and was at a ‘basic’ level. (Table 3 and Figure 3) The cumulative scores and component scores increased in all the four regional hospitals between 2019 and 2021 (Appendix A).

#### 3.2.2. Change in IPC Scores in District Hospitals from 2019 to 2021

In the district hospitals (N = 8), the overall IPC grade remained at ‘basic’ level. The highest increases in the percentage scores were seen in the ‘HAI surveillance’ component (45% change) and ‘IPC education and training’ (20% change). Less than 5% increase in the percentage scores were seen in the ‘workload, staffing and bed occupancy’ (3% change) and ‘built environment, materials and equipment for IPC at the facility level’ (4% change) components. (Table 3 and Figure 4) The cumulative scores and component scores increased in all the eight district hospitals between 2019 and 2021 (Appendix A).

#### 3.2.3. Gaps in the IPCAF Sub-Components at Regional and District Hospitals

The median percentage scores and gaps in the sub-components identified in 2021 at regional and district hospitals are shown in Table 4. Five sub-components in the regional hospitals and three sub-components in the district hospitals achieved 100% scores. Six sub-components had inadequate scores (≤25%) in the regional hospitals and eight had inadequate scores (≤25%) in the district hospitals. The sub-components with zero scores in both regional and district hospitals included: ‘a multidisciplinary team for implementing IPC multimodal strategies’; ‘a well-defined monitoring plan with clear goals, targets and activities’; and ‘senior facility leadership commitment and support for the IPC programme: by allocated budget specifically for the IPC activities’. Additionally, ‘assessment of hospital staffing needs’ had a zero score in the district hospitals. ‘Expertise in IPC/infectious disease to develop or adapt guidelines’ and “HAI surveillance performed” were also gaps in the district hospitals.

## 4. Discussion

This first country-wide assessment of IPC performance in the COVID era showed that IPC grades improved to ‘intermediate’ (51–75%) level at the national programme unit and at regional hospitals, but remained ‘basic’ (25–50%) at district hospitals. Although this reflects progress, the findings serve as an urgent call for a quantum jump in improving IPC performance, with a particular focus on district hospitals. The priority gaps at the national IPC unit were HAI surveillance, while at hospitals it was on assessment of hospital staffing needs and built environment (infrastructure), materials and equipment for IPC implementation.

The findings are important as they add contextual justification for the recent (2022) call by the WHO Director-General to prioritize IPC as a cornerstone of health system strengthening and universal health coverage [20]. Furthermore, optimal IPC implementation and monitoring is relevant to meeting the Sustainable Development Goals (targets 3.1 to 3.3 and 3.8, and those of Goal 6), especially that focused on reducing AMR (3.d.2) [21].

The recent revelation of five million annual global AMR deaths also re-emphasizes the importance of IPC [22]. The notion that ‘every infection prevented is an antibiotic treatment avoided, and possibly a death averted’ is now more relevant, particularly to Western sub-Saharan African countries such as Sierra Leone, which have a higher risk of AMR deaths.

The study had several strengths. First, it was country-wide involving simultaneous assessment of both the national IPC unit and the regional and district hospitals, and the findings are therefore likely to reflect operational realities and are generalizable. Second, the subject matter is an identified national operational research priority, which favours the potential for research uptake. Third, we used standardized WHO checklists and all assessments were carried out by the same external set of well-trained staff, thus limiting observer bias. Finally, we also adhered to the STROBE (Strengthening the Reporting of Observational Studies in Epidemiology) guidelines for the conduct and reporting of observational studies [23].

One of the study limitations was that we did not include private hospitals, which can be carried out in future research. Another limitation is that we did not include in-depth qualitative explorations to better understand the reasons for identified gaps. In future, mixed-methods research would be merited.

There are several policy and practice implications from the study findings. First, it is encouraging that Sierra Leone has a functional national IPC programme, which 55% of low-income countries were yet to achieve by 2018 [24]. The country also achieved a similar status at health facilities, which is in contrast with the observations reported from Ghana [25] and Uganda [16]. However, the overall progress in IPC implementation between 2019 and 2021 has been slower than desired. This may be explained by the fact that although the IPC programme in Sierra Leone is functioning within a clearly defined scope of responsibilities (policy, action plan, and framework for implementation) and dedicated professionals [11,26], it does not have a dedicated budget. Advocacy is needed to ensure dedicated resources for IPC at national and facility levels.

Second, an IPC guideline is now available in the COVID-19 era, with emphasis on the WHO core components, AMR and outbreak management. This document provides a framework for further improvement of the IPC programme and also galvanizes the development of health facility specific guidelines. The latter is an identified gap area due to lack of facility-level expertise (expertise currently at 13% in district hospitals and 50% in regional hospitals). The way forward would be to prioritize expert mentorship and dedicated time to facility-level IPC staff so that they understand and can implement the content of this guiding document. There is also a need to utilize the existing monitoring framework with clear goals, targets and activities in both regional and district hospitals (an identified gap).

Third, the national IPC programme unit conceived a national curriculum for IPC training and education (2020) and supported the training of at least one lead trainer per facility. These efforts reflected positively in the performance of “IPC education and training”, with regional hospitals reaching ‘advanced’ (83%) level and district hospitals moving to the doorstep of ‘advanced’ (75%) grades. This progress was catalysed by the trainings conducted during the initial phase of the COVID-19 pandemic and might explain why healthcare workers were found to be adherent to COVID-19 safe practices (regularly washing or sanitizing their hands and use of facemasks at point of care) [27]. However, a gap area on ‘monitoring the abilities of those trained to implement their acquired knowledge’ can be bridged through routine national level surveys on IPC practices and knowledge among healthcare workers.

Fourth, as in other low-income countries [24], the HAI surveillance remained at a ‘Basic’ grade in Sierra Leone. The national IPC programme was sucked into a ‘chicken and egg’ situation by formulating HAI surveillance objectives in the face of poor or non-existent access to microbiological laboratories—access to laboratories is an important requirement for surveillance. This was despite substantial improvements made in all the regional (40% increase) and district hospitals (45% increase) in surgical site infection (SSI) surveillance based on clinical assessments. This ongoing activity sets the pace for further improvement in HAI surveillance when access to microbiological laboratories becomes a reality through the Fleming fund grant to Sierra Leone [28].

Fifth, health care staff availability assessments according to the national norms on patient/staff ratios are important to optimize IPC implementation. Sierra Leone has a huge deficit in the health workforce (73%), worsened by the deaths of healthcare workers during the 2014–2015 EVD outbreak [29]. A previous study on IPC compliance also highlighted the deficiency of the healthcare workforce, including sanitary workers, and this needs to be addressed [17]. There is also a need to bridge the rhetoric of addressing the human resource gap by acquiring ‘hands-on deck’ to do the job.

Finally, the built environment (infrastructure) should have reached the ‘advanced’ level, but this has been a thorn in the flesh since the pre-COVID era [17]. Undeniably, IPC implementation is dependent on the availability of safe and sufficient water, adequate numbers of functional toilet facilities, facilities for sterilization, and waste disposal systems (e.g., functional incinerators). This also remains a challenge in other countries such as Kenya and Malawi [30,31]. Further, the availability of consumables such as soap, alcohol-based hand rub and Personal Protective Equipment (PPE) kits have remained scarce in Sierra Leone [27] as reported from several other settings [32,33].

The progress made in IPC program at national level also had a corresponding effect on hospital IPC programs. This will then give a good recommendation for countries with similar health system to strengthen the national IPC program objectives and descend at subnational level.

## 5. Conclusions

Although there has been encouraging improvements in IPC implementation country-wide, the progress between 2019 and 2021 has been slower than desired, with the current overall status ranging from ‘basic’ to ‘intermediate’. In light of the current COVID-19 pandemic and endemic infectious disease outbreaks in Sierra Leone, a more rapid pace of progress is needed in IPC implementation. This will depend on successful mobilization of political will, leadership, resources and accountability. It is high-time to take that leap forward.

## Figures and Tables

**Figure 1 ijerph-19-05642-f001:**
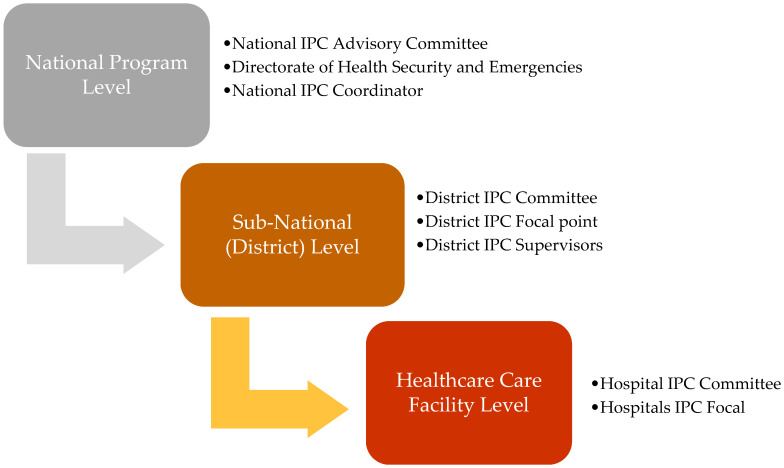
Organizational structure of IPC program in Sierra Leone.

**Figure 2 ijerph-19-05642-f002:**
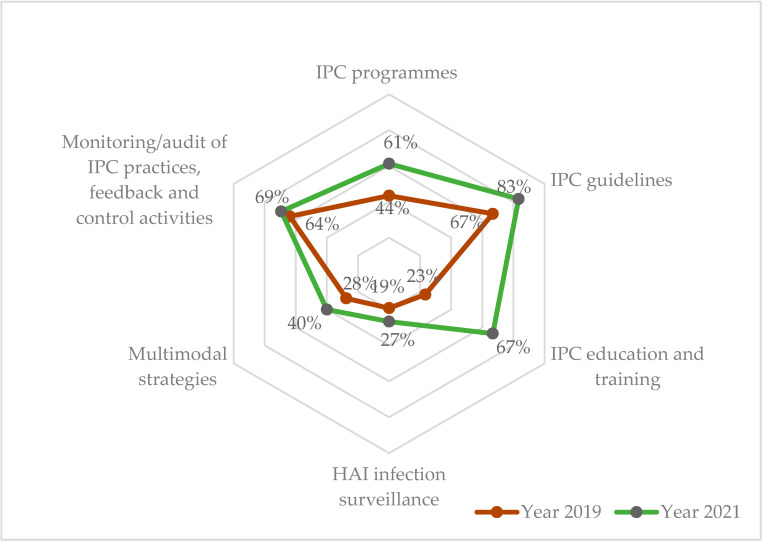
Radar chart showing IPC compliance scores at National IPC unit assessment in 2019 (pre-COVID-19) and 2021 (COVID-19 era) in Sierra Leone. Note: The radar chart shows the IPC component score emanating from the centre (0%) expanding outwards to a maximum of 100%. The 2021 scores (green line) lie outside the 2019 scores (red line), indicating overall improvement in all the components of IPC. Abbreviation: IPC = Infection, Prevention and Control; HAI = Healthcare Associated Infection.

**Figure 3 ijerph-19-05642-f003:**
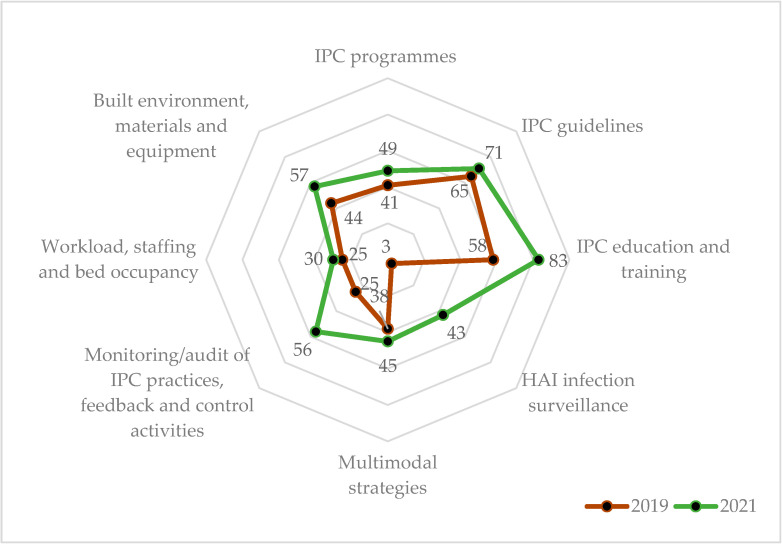
Median IPC compliance percentage scores across eight core components in the four regional hospitals during 2019 (pre-COVID-19) and 2021 (COVID-19 era), Sierra Leone. Note: The radar chart shows the IPC component score emanating from the centre (0%) expanding outwards to a maximum of 100%. The 2021 scores (green line) lie outside the 2019 scores (red line), indicating overall improvement in all the components of IPC. Abbreviation: IPC = Infection, Prevention and Control; HAI = Healthcare Associated Infection.

**Figure 4 ijerph-19-05642-f004:**
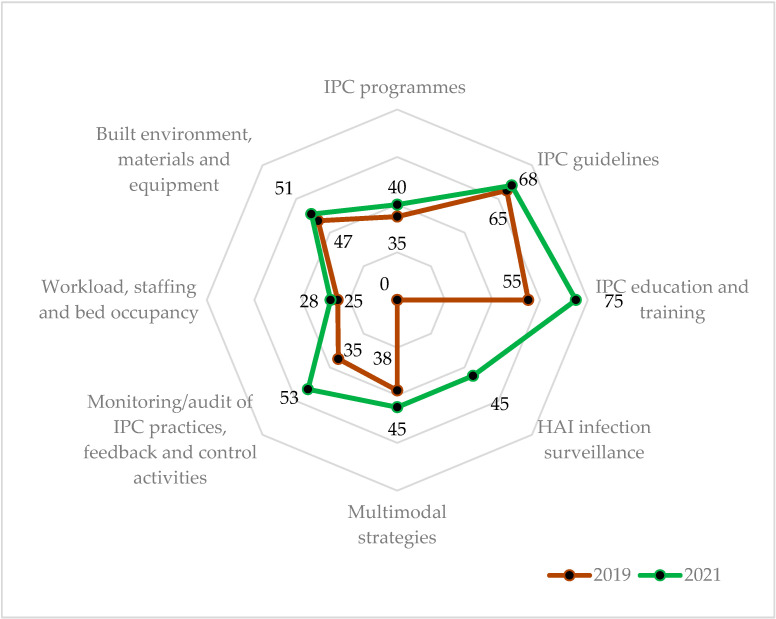
Median IPC compliance percentage scores across eight core components in the eight district hospitals during 2019 (pre-COVID-19) and 2021 (COVID-19 era), Sierra Leone. Note: The radar chart shows the IPC component score emanating from the centre (0%) expanding outwards to a maximum of 100%. The 2021 scores (green line) lie outside the 2019 scores (red line), indicating overall improvement in all the components of IPC. Abbreviation: IPC = Infection, Prevention and Control; HAI = Healthcare Associated Infection.

**Table 1 ijerph-19-05642-t001:** Percentage change in minimum IPC score at the national IPC unit between 2019 (pre-COVID-19) and 2021 (COVID-19 era) in Sierra Leone.

IPC Core Components ^a^	2019			2021			% Change ^d^
	Grade ^b^	Score	(%) ^c^	Grade ^b^	Score	(%) ^c^	
i. IPC programme	Basic	44	(44)	Intermediate	61	(61)	17
ii. IPC guidelines	Intermediate	67	(67)	Advanced	83	(83)	16
iii. IPC education and training	Inadequate	23	(23)	Intermediate	67	(67)	44
iv. HAI surveillance	Inadequate	19	(19)	Basic	27	(27)	8
v. Multimodal strategies	Basic	28	(28)	Basic	40	(40)	12
vi. Monitoring/audits of IPC practices and feedback	Intermediate	64	(64)	Intermediate	69	(69)	5
Cumulative score (%)	Basic	245	(41)	Intermediate	347	(58)	17

Abbreviation: IPC = Infection, Prevention and Control; HAI = Healthcare Associated Infection. ^a^ Maximum score for each component is 100 and for the cumulative it is 600. ^b^ Grade: IPC performance in each component is graded based on the obtained percentage: (i) inadequate (0–25%), (ii) basic (25.1–50%), (iii) intermediate (50.1–75%), and (iv) advanced (75.1–100%). ^c^ Percentages are calculated relative to the maximum score for the component. ^d^ Percentage in 2021–Percentage in 2019.

**Table 2 ijerph-19-05642-t002:** The percentage scores of the sub-components of IPCAT at the national IPC unit of Sierra Leone during 2019 (pre-COVID-19) and 2021 (COVID-19 era) assessment.

IPC Core Components	Sub-Components	2019	2021
i. IPC Programme	Organization and leadership of the programme	63%	63%
	Defined scope of responsibilities	43%	71%
	Linkages with other programmes and professional organizations	25%	50%
ii. IPC Guideline	Development, dissemination, and implementation of national technical guidelines	67%	100%
	Education and training of relevant healthcare workers on IPC guidelines	33%	67%
	Monitoring of guideline adherence	100%	100%
iii. IPC education and training	Supporting and facilitating IPC education and training at the facility level	60%	100%
	National curricula and IPC training and education	0%	100%
	Monitoring of training and education	0%	0%
	Implementation of training and education	33%	67%
iv. HAI surveillance	Coordination of surveillance at the national level	29%	43%
	National objectives of surveillance	20%	20%
	Prioritized HAIs for surveillance	0%	17%
	Methods of surveillance	67%	67%
v. Multimodal strategies	National and sub-national coordination in support of local implementation of IPC improvement interventions	50%	100%
	National and sub-national facilitation in support of local implementation of IPC improvement interventions	60%	60%
	Programme and accreditation linkages	0%	0%
vi. Monitoring/audits of IPC practices and feedback	Monitoring/audit and feedback framework for IPC	50%	50%
	Monitoring/audit indicators	75%	75%
	Monitoring/audit and feedback process and reporting	67%	83%

Abbreviation: IPC = Infection, Prevention and Control; HAI = Healthcare Associated Infection. 
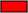
 Inadequate; 
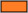
 Basic; 
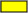
 Intermediate; 
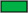
 Adequate.

**Table 3 ijerph-19-05642-t003:** Percentage change in median IPC scores at regional (N = 4) and district hospitals (N = 8) between 2019 (pre-COVID-19) and 2021 (COVID-19 era) in Sierra Leone.

IPC Core Components ^a^	RH (N = 4)					DH (N = 8)				
	2019		2021		% Change ^d^	2019		2021		% Change ^d^
	Score	(%) ^b^	Score	(%) ^b^		Score	(%) ^b^	Score	(%) ^b^	
i. IPC programme	41	(41)	49	(49)	8	35	(35)	40	(40)	5
ii. IPC guidelines	65	(65)	71	(71)	6	65	(65)	68	(68)	3
iii. IPC education and training	58	(58)	83	(83)	25	55	(55)	75	(75)	20
iv. HAI surveillance	3	(3)	43	(43)	40	0	(0)	45	(45)	45
v. Multimodal strategies	38	(38)	45	(45)	8	38	(38)	45	(45)	8
vi. Monitoring/audits of IPC practices and feedback	25	(25)	56	(56)	31	35	(35)	53	(53)	18
vii. Workload, staffing and bed occupancy	25	(25)	30	(30)	5	25	(25)	28	(28)	3
viii. Built environment, materials and equipment for IPC at the facility level	44	(44)	57	(57)	13	47	(47)	51	(51)	4
Cumulative score (%)	299	(37)	434	(54)	17	300	(37)	405	(50)	13
Grading ^c^	Basic		Intermediate			Basic		Basic		

Abbreviation: IPC = Infection, Prevention and Control; RH = regional hospital; DH = district hospital; HAI = Healthcare Associated Infection. ^a^ Maximum score for each component is 100 and for the cumulative it is 800. ^b^ Percentages are calculated relative to the maximum score for the component. ^c^ Grade: IPC performance in each component will be graded based on the obtained percentage: (i) inadequate (0–25%), (ii) basic (25.1–50%), (iii) intermediate (50.1–75%), and (iv) advanced (75.1–100%). ^d^ Percentage in 2021–Percentage in 2019.

**Table 4 ijerph-19-05642-t004:** The percentage scores of the sub-components of IPCAF at the regional and district hospitals of Sierra Leone in 2021.

IPC Core Components	Sub-Components	Median Percentage Score *	
		RH (N = 4)	DH (N = 8)
i. IPC Programme	IPC programme at facility	50%	50%
	Functional IPC committee	100%	50%
	Senior facility leadership commitment and support for the IPC programme: by allocated budget specifically for the IPC activities	0%	0%
ii. IPC Guideline	Expertise in IPC to develop or adapt guidelines	50%	13%
	Availability of IPC guidelines	57%	56%
	Consistent with national/international guidelines	100%	100%
iii. IPC education and training	Availability of personnel with the IPC expertise to lead IPC training	100%	100%
	Frequency of IPC training	83%	50%
	IPC training integrated in the clinical practice and training of other specialties	13%	13%
iv. HAI surveillance	Surveillance as a defined component of IPC programme	100%	75%
	HAI surveillance performed	14%	11%
	Methods of surveillance	45%	20%
v. Multimodal strategies	Use of Multimodal strategies to implement IPC interventions	100%	100%
	Multimodal strategies elements implemented in an integrated way	40%	40%
	A multidisciplinary team for implementing IPC multimodal strategies	0%	0%
vi. Monitoring/audits of IPC practices and feedback	A well-defined monitoring plan with clear goals, targets and activities	0%	0%
	Monitoring of IPC processes and indicators	50%	32%
	Feedback of auditing reports on the state of the IPC activities/performance	60%	63%
vii. Workload, staffing and bed occupancy	Assessment of hospital staffing needs	25%	0%
	Hospital bed occupancy	64%	44%
viii. Built environment, materials and equipment for IPC at the facility level	Water availability and access	58%	40%
	Functioning Hand hygiene and sanitation facilities	50%	65%
	Patient placement and personal protective equipment (PPE) in health care settings	58%	50%
	Medical waste management, and sewage	54%	49%
	Decontamination and sterilization	42%	50%

Abbreviation: IPC = Infection, Prevention and Control; HAI = Healthcare Associated Infection, RH = Regional Hospital, DH = District Hospital. * The % score for each component in the facility was calculated using IPCAF scores. The median of this score among regional hospitals and district hospitals is presented. 
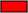
 Inadequate; 
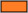
 Basic; 
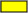
 Intermediate; 
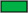
 Adequate.

## Data Availability

The metadata record of the data used in this paper is available at DOI https://doi.org/10.6084/m9.figshare.19134608 (accessed on 28 January 2022). Requests to access these data should be sent to the corresponding author.

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
