# Peer review of "Achieving Minimum Standards for Infection Prevention and Control in Sierra Leone: Urgent Need for a Quantum Leap in Progress in the COVID-19 Era!"

_ijerph, 2022, doi:10.3390/ijerph19095642_

Round 1
Reviewer 1 Report
This study examines the change in Infection prevention and control (IPC) measures in Sierra Leone pre- and during-Covid-19. It reports that there is some improvement, though it is less than the desired levels. Overall the paper is well written and examines a significant issue dealing with healthcare staff. However, I have some comments as given below.
1) Authors may shed more light on the governance structure of national and lowever levels hospitals. Who oversee these all these hospitals? Who is responsible for the gap in performance in lower level hospitals?
2) How you ensure that IPC measures given in records are actually being followed? No doubt the they were more likely to be followed during Covid-19 because of the fear.
3) Do you think there is trade-off between IPC measures and the service provided to the patients infected with severe viral diseases, as the Covid-19.
4) Give some implications to improve governance structure of hospitals to ensure IPC implementation.
5) What other countries with similar health systems can learn from this study?
Author Response
Reviewer 1 general remarks: This study examines the change in Infection prevention and control (IPC) measures in Sierra Leone pre and during-Covid-19. It reports that there is some improvement, though it is less than the desired levels. Overall the paper is well-written and examines a significant issue dealing with healthcare staff. However, I have some comments as given below.
Point 1. Authors may shed more light on the governance structure of national and lower levels hospitals. Who oversee these all these hospitals? Who is responsible for the gap in performance in lower level hospitals?
Response for Point 1
Thank you very much for this valuable comment. Part of this was addressed in the specific settings area of the methods section. However, additional inputs have been made in lines 152-158 and Figure 1 added to fully address this point.
Point 2. How you ensure that IPC measures given in records are actually being followed? No doubt they were more likely to be followed during Covid-19 because of the fear.
Response for Point 2
Thanks for shedding more light on this. The IPCAT and IPCAF tools were designed to assess the compliance to IPC standards and not the actual practices of healthcare workers. We agree that some of the practices might have improved not only because of COVID-19 fear, but also due to the trainings conducted during the initial phase of the COVID-19 pandemic. A study by Kanu et al. reported higher adherebce to COVID-19 safe practices (regularly washing or sanitizing their hands and use of facemasks at point of care) in lines 371-374.
Point 3. Do you think there is trade-off between IPC measures and the service provided to the patients infected with severe viral diseases, as the Covid-19.
Response for Point 3
Thank you for the comment. This study was not designed to assess the effect of IPC measures on healthcare provision. However, we believe that adherence to IPC measures ensures safer care for patients. In countries like Sierra leone, with limited healthcare workforce, good IPC compliance also enables safe environment for healthcare providers, especially during outbreaks. This has been discussed in lines 387-393.
Point 4. Give some implications to improve governance structure of hospitals to ensure IPC implementation.
Response for Point 4.
Thank you for the comment. The IPC programme in Sierra Leone is functioning within a clearly defined scope of responsibilities (policy, action plan, and framework for implementation) and dedicated professionals. The IPC committees were established in all health facilities. Thus, we discussed on the requirements of makeing this committees functional in terms of having a dedicated budget in lines 347-356.
Point 5. What other countries with similar health systems can learn from this study?
Response for Point 5
We found this to be a very useful comment. We have nicely incorporated this point in lines 402-405
Reviewer 2 Report
Dear Authors,
The problem of communicable diseases prevention is currently very relevant. I appreciate taking up this topic, but in my opinion, the article has some flaws that should be corrected.
- Introduction – I suggest adding a part presenting a healthcare system in Sierra Leone briefly, its main problems and imitation, and describing how it deals with the pandemic. It would be interesting for readers who are not familiar with it and allowing to understand general challenges
- Apart from the introduction, I propose adding a section with a literature review presenting a disease prevention problem with particular emphasis on developing countries. It would be helpful to combine it with the issues related to the COVID pandemic
- In section 2 – “Materials and Methods”, there are too many levels of numbers which create unnecessary too short paragraphs
- The discussion is not a discussion. Results should be compared with other research in this area (especially from developing countries). I also suggest at least to try explaining the changes in IPC level using data describing overall hospital performance
- Title – I propose shortening it. The reference to COVID is illegitimate since the authors do not touch this problem
- Some keywords should be removed or changed – they misguide. Like “Antimicrobial resistance” which is not the important strand in this paper
- The paper needs additional editing (typing errors)
Author Response
Reviewer 2 general remarks: The problem of communicable diseases prevention is currently very relevant. I appreciate taking up this topic, but in my opinion, the article has some flaws that should be corrected.
Point 1. Introduction – I suggest adding a part presenting a healthcare system in Sierra Leone briefly, its main problems and imitation, and describing how it deals with the pandemic. It would be interesting for readers who are not familiar with it and allowing to understand general challenges.
Response for Point 1
Thank you for the comment. The healthcare system in Sierra Leone was described in the study setting. However, we have made an additional statement in 128-133 to fully address this point 1.
Point 2 . Apart from the introduction, I propose adding a section with a literature review presenting a disease prevention problem with particular emphasis on developing countries. It would be helpful to combine it with the issues related to the COVID pandemic.
Response for Point 2
Thank you for the suggestion. We believe that infectious disease prevention problem with particular emphasis on Sierra Leone has been highlighted in the introduction section (lines 69-74). As there is no separate section for literature review in the journal template, we tried to accommodate this in the introduction.
Point 3. In section 2 – “Materials and Methods”, there are too many levels of numbers which create unnecessary too short paragraphs
Response for Point 3
Thank you very much for your comment. We have improved on this and reduced the number of paragraphs. However, the structure and subheadings are per journal requirements and according to STROBE guideline we adhere.
Point 4. The discussion is not a discussion. Results should be compared with other research in this area (especially from developing countries). I also suggest at least to try explaining the changes in IPC level using data describing overall hospital performance
Response for Point 4
Many thanks for this comment. Throughout the discussion, we try to compare our study with other IPC studies mostly from Africa. To our knowledge, this is the only study that has assessed progress in IPC at both the national and hospital levels. A Pubmed search did not identify any studies that assessed IPC program at a national and health facility levels, also comparing IPC performance between pre-COVID-19 and COVID-19 periods.
Point 5. Title – I propose shortening it. The reference to COVID is illegitimate since the authors do not touch this problem.
Response for Point 5
Thanks for the suggestion. However, we would prefer to keep COVID-19 in the title for two reasons: first, it indicates the study period; second, it highlights one of main study findings, that even during COVID-19 the desired progress in IPC core components was not achieved, emphasizing the need for improvement.
Point 6. Some keywords should be removed or changed – they misguide. Like “Antimicrobial resistance” which is not the important strand in this paper
Response for Point 6
Thank you for the suggestion. This has been amended.
Point 7. The paper needs additional editing (typing errors)
Response for Point 7
Thanks for your valuable observation. Additional editing has been done and typing errors resolved.
Round 2
Reviewer 2 Report
Dear Authors,
I still believe that the work with existing literature is crucial from the scientific research point of view. If you do not want to add the new section, the literature review must be extended in the section “Introduction”.
According to the section “Discussion”, it is crucial to add it. You do not have to discuss the results strictly with the papers concerning IPC on two levels and pre and post covid. The discussion can be divided into several dimensions.
Author Response
Reviewer 2 Round 2 Point
I still believe that the work with existing literature is crucial from the scientific research point of view. If you do not want to add the new section, the literature review must be extended in the section “Introduction”. According to the section “Discussion”, it is crucial to add it. You do not have to discuss the results strictly with the papers concerning IPC on two levels and pre and post covid. The discussion can be divided into several dimensions.
Round 2 Response
Thank you, the suggested additions are made in the Introduction (Line numbers 57- 64).
In the discussion, under the policy and practice implications section, we compare our findings with other studies from Sub-Saharan African countries, which mimic the current study setting. We have tried to detail on the certain IPC components, which have direct implication on policy and practice in Sierra Leone, Ghana, Uganda, and other sub-Sharan countries (line numbers 354-355, 383-384, 399-406).